# Attitudes towards the Tick-Borne Encephalitis Vaccine among Children’s Guardians: A Cross-Sectional Survey Study in Poland

**DOI:** 10.3390/vaccines12080918

**Published:** 2024-08-15

**Authors:** Furkan Ates, Marta Dyszkiewicz, Julia Witkiewicz, Kacper Toczylowski, Dawid Lewandowski, Artur Sulik

**Affiliations:** Department of Pediatric Infectious Diseases, Medical University of Bialystok, Waszyngtona 17, 15-274 Bialystok, Poland; 39164@student.umb.edu.pl (F.A.); martadyszkiewicz1@gmail.com (M.D.); dawid.lewandowski@umb.edu.pl (D.L.); artur.sulik@umb.edu.pl (A.S.)

**Keywords:** tick-borne encephalitis, vaccination, survey, attitudes, risk perception, health knowledge, pediatrics

## Abstract

This cross-sectional survey study aimed to assess the attitudes and knowledge of children’s guardians in Poland towards tick-borne encephalitis (TBE) and its vaccine, as well as to compare these attitudes to those regarding other infectious diseases and their vaccines. Data were collected through anonymous questionnaires, both paper-based and online, from 3030 respondents across Poland, with the majority being from TBE-endemic areas. The survey included questions on demographic characteristics, general vaccination beliefs, and specific perceptions of TBE and its vaccine. Statistical analysis revealed significant associations between willingness to vaccinate against TBE and factors such as general vaccination attitudes, information sources, vaccine safety ratings, and perceptions of disease severity. Results indicated that guardians from larger cities and those with fewer children were more likely to vaccinate. Additionally, parents who supported general vaccinations were significantly more willing to vaccinate against TBE. The study concludes that enhancing public awareness about the safety and importance of the TBE vaccine, especially in endemic regions, is crucial for improving vaccination rates. Targeted public health interventions addressing misconceptions and providing accurate information are essential strategies to increase TBE vaccine uptake and protect children from this serious disease.

## 1. Introduction

Tick-borne encephalitis (TBE) is a potentially serious viral infectious disease involving the central nervous system [1,2,3]. The causative virus bearing the same name—tick-borne encephalitis virus or TBE-virus (TBEV) in short—has three distinct subspecies, but essential for this study is the Central European encephalitis subtype, also referred to as the Western subtype in general [4,5]. The vectors of this virus belong to the *Ixodes* species—in Europe, it is transmitted by *Ixodes ricinus* [6]. Today, the most effective prevention and prophylaxis method for TBE is vaccination [7,8,9]. With the number of TBE cases increasing in many countries including Poland, which poses a concern to the healthcare system, vaccinating people for the disease has become more important than ever [4,10]. The voivodeship Podlaskie in Northeastern Poland in particular has always been an endemic area for TBE and other tick-borne diseases. However, little is known about tick-borne disease awareness in general and TBE-vaccination rates remain low in the EU/EEA since it is not part of the obligatory national vaccination schedule in many countries [11].

To be more precise, in 2022, Poland reported higher vaccination rates for several recommended vaccines like the influenza vaccine among all age groups (1,107,351 shots given in total) and the COVID-19 vaccine among individuals aged 0–17 years (1,007,299 shots given in total) [12]. Compared to other recommended vaccines, the coverage for the chickenpox and meningococcal vaccines had a moderate uptake, with 123,436 and 126,081 shots given in total in 2022, respectively [12].

In contrast, the coverage rates for other vaccines were notably lower. Only 57,830 doses of the human papillomavirus (HPV) vaccine and 36,127 doses of the hepatitis A (HAV) vaccine were administered [12]. Also concerning is the low uptake of the tick-borne encephalitis (TBE) vaccine, with just 83,020 doses given [12].

While considerable research has advanced our understanding of how parents perceive other infectious diseases or vaccines, like Lyme borreliosis [11] and what parental practices, and prior experiences related to TBE there are [13], a critical research gap revolves around an extensive analysis of children’s guardians’ attitudes and perception of TBE and the TBE vaccine and comparing it to other infectious diseases and their respective vaccines in Poland.

Assessment of attitude, perception, and knowledge about TBE and its vaccine is of fundamental importance when learning about factors influencing parental willingness to uptake the TBE vaccine for their children. Ultimately, this way misinformation can be directly targeted, and accurate education ensured. This is the objective for increasing vaccination rates, which is crucial for preventing further spread of TBE and reducing the burden of the disease on individuals.

Thus, the aim of our study is to assess the attitudes and knowledge of children’s guardians in Poland to TBE and its vaccine as well as to compare it to other infectious diseases and their respective vaccines. The significance of this research lies in its potential to uncover the factors influencing TBE-vaccine acceptance and hesitancy, which can inform public health strategies to improve vaccination coverage. The hypotheses tested include whether parents’ demographics, general support for vaccinations, and used information sources are related with their willingness to vaccinate against TBE and how perceptions of vaccine safety and disease severity impact their decisions.

## 2. Materials and Methods

### 2.1. Study Design and Setting

We conducted a cross-sectional study among the parents and guardians of children based on an anonymous questionnaire that contained 17 questions. Paper survey was administered at the Department of Pediatric Infectious Diseases at the Medical University of Bialystok, Poland. The questions were answered by parents of children hospitalized in the above-mentioned ward. All of them lived in northeastern Poland, an area specific for the occurrence of TBE. Online survey was shared in several Facebook groups and filled out by responders from all over Poland. Data was collected from March 2020 to June 2022. The study was approved by the Ethics Committee of the Medical University in Bialystok (approval number APK.002.30.2020).

The collected 3030 surveys analyzed attitudes towards the TBE vaccine compared to other recommended vaccinations.

We assessed guardians’ attitudes toward vaccinations, their safety, knowledge of methods of enhancing immunity, attitudes toward the effectiveness of vaccinations as a method of prevention, vaccinating a child with non-mandatory vaccine, free access to highly combined vaccines, knowledge of the side effects of vaccinations and problems in discussing vaccination with a doctor; by using multiple choice questions.

In two questions: “Question 10: How much of a health threat do you consider the following infectious diseases to be?” and “Question 11: How safe do you think the following vaccines are?” participants responded via the five-point Likert scale, ranging from 1—“not dangerous” to 5—“very dangerous” for Question 10 and from 1—“not safe” to 5—“very safe” for Question 11.

In “Question 8: Where do you get your knowledge about vaccinations?”, a distinction is drawn between reliable and non-reliable categories based on the credibility and authority typically associated with the sources. Reliable sources in this analysis include medical personnel such as doctors, pharmacists, and nurses, alongside authoritative health organizations like the WHO and CDC. These sources are regarded as reliable due to their foundation in scientific evidence, regulatory oversight, and professional expertise, which are crucial for disseminating accurate health information. Conversely, all other sources not fitting these criteria are classified as non-reliable.

The full questionnaire is available in the Appendix A.

### 2.2. Information about the Interviewees

Data concerning sociodemographic values (gender, age, education, size of hometown, number of children) were analyzed. Respondents were asked whether they vaccinated or plan to vaccinate their child against TBE and if they consider TBE to be a threatening disease to their health, and whether they consider the TBE vaccine to be safe.

### 2.3. Data Analysis

Data were collected through anonymous questionnaires printed on paper and questionnaires filled out by interested users of Facebook groups. We collected 3030 questionnaires in total—2543 online and 487 in paper.

We ensured the quality of this survey by conducting two small pilot phases, each with a sample size of 20. During these phases, the initial version of the survey was distributed among the hospital staff, students and hospitalized parents, and their feedback was collected to improve the survey’s quality. The final version of the questionnaire was prepared based on this feedback. To minimize errors, two independent individuals entered and compared the paper-based questionnaires using Microsoft Excel. The principal investigator reviewed vague responses to determine the correct answer. Surveys that were illegible or had the majority of answers missing were removed. If only a few key answers were missing, we included the questionnaire but accounted for missing data during analysis. Prior to the analysis, we checked each question to ensure the proportion of missing data was below 5%. From the paper questionnaires we had to delete 6.7% of questionnaires.

The alpha level (α) was set at 0.05, indicating that *p*-values falling below this cutoff are deemed to reflect statistically significant differences. For categorical variables, distributions were characterized by counting occurrences (n) and computing the respective percentages (%) for each category. The assessment of statistical significance regarding differences between two independent groups concerning categorical variables was performed using Pearson’s Chi-squared test, Fisher’s exact test, and a test for proportions.

Analyses were conducted using the R Statistical language (version 4.3.1; R Core Team, 2023) on Windows 10 pro 64 bit (build 19045), using the packages report (version 0.5.7), gtsummary (version 1.7.2), dplyr (version 1.1.3), and ggplot2 (version 3.4.4).

## 3. Results

### 3.1. Characteristics of the Study Group

The demographic characteristics of the participants are detailed in Table 1, which presents the overall profile for the total sample and is segmented by the participants’ willingness to vaccinate their children against TBE.

Female representation was predominant (91.24%), with mothers actively involved in children’s healthcare decisions. The age group 30–39 was most common (60.05%), showing diverse parental ages considering TBE vaccination. High education levels were prevalent (78.07%). Participants were mainly from large cities (35.53%) and smaller cities (26.68%), with rural areas and small towns forming a smaller fraction. Most parents had 1–2 children (78.15%), while some had none (2.29%) or three or more (19.56%). A significant portion (30.96%) came from TBE-endemic areas, which underscores the regional relevance of its vaccination.

These data reflect closely but not directly the demographics of Polish society. Female representation in Polish society is 51.7%, and the median age in Poland is 41.7. Higher education is held by 24.5% individuals aged 13 and above. Urban areas in Poland are inhabited by 59.8% of the total population [14].

### 3.2. Parental Demographic Profiles Stratified by Willingness to Vaccinate against TBE

Gender may not be a strong predictor of vaccination willingness among parents (*p* = 0.087). Age group analysis indicates specific trends: very young parents (up to 19 years old) and those between 50–59 years exhibit a higher willingness to vaccinate, with significant *p*-values of 0.027 and 0.045, respectively.

In terms of education, the analysis indicates no significant differences across educational levels (*p* = 0.355).

Locality proves to be a significant factor, with parents from larger cities (over 300,000 inhabitants) showing a markedly higher willingness to vaccinate (40.51% vs. 33.53%, *p* < 0.001).

The number of children in a family significantly impacts vaccination decisions. Parents with no children are much more likely to vaccinate (5.33% vs. 1.07%, *p* < 0.001). Conversely, those with three or more children show a lower willingness (14.60% vs. 21.54%, *p* < 0.001).

Finally, the voivodeship endemic for TBE shows a significant effect on vaccination willingness (35.32% vs. 29.19%, *p* = 0.001).

### 3.3. Parental General Attitudes towards Vaccination

In efforts to understand the broader societal attitudes towards vaccinations and their direct influence on specific health decisions, this section presents a comparative analysis focused on general vaccination beliefs and the willingness of parents to vaccinate their children against TBE. Table 2 below illustrates the dependencies between general attitudes towards vaccination and decision-making regarding the TBE vaccine for children.

72.57% of surveyed individuals support child vaccination, showing strong general backing for vaccinations. Only 27.38% see vaccines as riskier compared to not vaccinating their child, with 68.53% recognizing the greater risk of not vaccinating. Just 27.62% prefer natural methods over vaccination for infection prevention, indicating trust in vaccinations. The data also highlights economic considerations impacting vaccination decisions. When presented with the hypothetical scenario of free access to combined vaccines, which are typically more expensive and not covered by public healthcare in Poland, 57.64% of participants expressed willingness to use them for their children. Lastly, concerns about vaccines causing autism persist (27.10%) despite extensive scientific evidence disproving the link.

#### Parental General Attitudes towards Vaccination Stratified by Willingness to Vaccinate against TBE

A striking 95.98% of respondents who support general vaccinations for children also express willingness to vaccinate against TBE, compared to a mere 63.11% who do not support TBE vaccinations, highlighting a strong alignment between general pro-vaccination attitudes and specific vaccination actions. The disparity is more pronounced among those opposed to general vaccinations, with only 1.72% willing to vaccinate against TBE, indicating that negative perceptions of vaccinations significantly deter specific vaccine uptake.

Concerns about the dangers of vaccinating versus not vaccinating children show that a majority view the lack of vaccination as more hazardous (91.72% of those willing to vaccinate against TBE versus 59.18% who are not).

Conversely, only a small fraction (5.52%) who consider vaccinations dangerous are willing to vaccinate, reinforcing the idea that safety concerns are a critical barrier to vaccination.

Beliefs in natural immunity over vaccinations also affect TBE vaccination decisions. A mere 4.60% of respondents who prioritize natural methods are willing to vaccinate against TBE, compared to 82.39% who favor vaccinations over natural immunity. This indicates a strong belief in the efficacy of vaccinations over natural methods among those choosing to vaccinate.

Economic factors are also pivotal. The data reveals that if high-combination vaccines were freely available, 81.63% of parents willing to vaccinate against TBE would opt to use them, compared to only 47.96% of parents unwilling to vaccinate against TBE. This suggests that financial barriers significantly affect vaccination decisions, and removing these could enhance vaccine uptake.

Finally, the persistent myth connecting vaccinations to autism significantly influences decisions, with only 7.58% of those who believe in this link willing to vaccinate against TBE. This is in stark contrast to 71.41% who do not believe in the link and are willing to vaccinate, illustrating that misinformation continues to hinder vaccination efforts.

### 3.4. Parental Attitudes toward Non-Mandatory Childhood Vaccinations

In the contemporary vaccine landscape, parents face decisions not only about mandatory vaccinations but also about optional or non-mandatory vaccines that are not included in the standard vaccination schedule. Understanding parental attitudes towards these non-mandatory vaccinations provides insight into public health trends and can help guide policy and educational strategies.

The following analysis explores how these attitudes correlate with the willingness of parents to vaccinate their children against TBE, a disease also not typically included in the primary vaccination schedule. Appendix A show that the willingness to vaccinate against TBE, at 28.78%, while substantial, is significantly less pronounced (*p* < 0.001) when compared to other non-mandatory vaccinations such as chickenpox and meningococcal disease, which show higher acceptance rates of 36.71% and 38.78%, respectively. In contrast, influenza (17.20%) and hepatitis A (20.00%) vaccinations have lower willingness, possibly due to seasonal factors or perceived risks. HPV vaccination willingness rates (23.10%) are also moderate, reflecting ongoing education efforts. A notable 39.67% opted not to vaccinate against any non-mandatory diseases, indicating diverse factors like vaccine skepticism or perceived low risk.

#### Parental Attitudes toward Non-Mandatory Childhood Vaccinations Stratified by Willingness to Vaccinate against TBE

This difference is statistically significant across all listed diseases, indicating a strong correlation between the willingness to vaccinate against TBE and the acceptance of other non-mandatory vaccines.

Figure 1 shows that for chickenpox, a substantial 63.88% of parents willing to vaccinate against TBE also chose to vaccinate their children against chickenpox, compared to only 25.73% among those not willing to vaccinate against TBE.

The pattern is similar for other vaccines. For instance, the willingness to vaccinate against influenza is 36.93% among TBE-vaccinating parents, significantly higher than the 9.23% among those who are not, which underscores a consistent trend where acceptance of one non-mandatory vaccine enhances the likelihood of accepting others.

The meningococcal vaccine shows the highest acceptance among TBE-vaccine-willing parents at 67.66%, compared to 27.11% among those unwilling, which not only highlights a proactive approach towards vaccines considered important but also mirrors public health priorities regarding severe bacterial infections.

The acceptance rates for hepatitis A and HPV vaccines stand at 46.67% and 46.90%, respectively, among parents willing to vaccinate against TBE, as opposed to markedly lower rates of 9.22% and 13.48% among those who are not. These figures suggest that educational and awareness campaigns targeting one vaccine might inadvertently raise awareness and acceptance of other vaccines.

The acceptance rates for hepatitis A and HPV vaccines stand at 46.67% and 46.90%, respectively, among parents willing to vaccinate against TBE, as opposed to markedly lower rates of 9.22% and 13.48% among those who are not.

Most strikingly, the category “None of the above” is selected by only 0.23% of those willing to vaccinate against TBE but jumps to 55.61% among those who are not, indicating a profound divide in general openness to vaccinations between the two groups.

### 3.5. Information Sources on Vaccinations Stratified by Willingness to Vaccinate against TBE

In the pursuit of understanding where parents acquire their knowledge about vaccinations, particularly in relation to their willingness to vaccinate their children against TBE, a detailed survey was conducted. The results, which are visualized in Figure 2 and Appendix A, delineate various sources from which parents might gather information, ranging from healthcare professionals to digital and traditional media. The analysis further explores how these sources influence the decision-making process of those who chose to vaccinate against TBE compared to those who did not. Herein, we provide a comparative analysis of these findings to better understand the impact of different information channels on vaccination decisions.

A higher proportion of parents who are willing to vaccinate their children against TBE tend to consult doctors, with 72.71% of them doing so compared to 66.48% of those who are not willing, a difference that is statistically significant (*p* = 0.001).

The role of the CDC and WHO as trusted sources of information is also pronounced; 48.74% of those willing to vaccinate rely on these organizations, compared to only 39.85% of those unwilling, with the difference being statistically significant (*p* < 0.001).

Conversely, the Internet shows an interesting trend where a higher percentage of those unwilling to vaccinate (66.13%) consult this medium compared to 60.21% of those willing, with this difference being statistically significant (*p* = 0.002).

Sources like books and guides, TV, and non-medical journals show a significant divergence in consultation rates between those willing and those unwilling to vaccinate. Notably, books and guides are consulted less by those willing to vaccinate (16.40%) compared to those unwilling (26.83%), suggesting that detailed or possibly outdated written resources might contain information that dissuades vaccination. Similarly, non-medical journals are consulted more by those unwilling to vaccinate, further underscoring the potential influence of less regulated sources in shaping negative vaccination attitudes.

Other sources, such as pharmacists, nurses, friends, social media, and posters and flyers, show no significant difference in consultation rates between the two groups or are less influential overall. This highlights that while these sources are used, they do not distinctly influence vaccination decisions in the context of TBE to the same extent as doctors, authoritative health organizations, and the Internet.

#### An Assessment of the Information Sources in Terms of the Reliability

The data in Table 3 clearly demonstrate that access to and the choice of information between reliable and non-reliable sources significantly affect parental decisions regarding TBE vaccination.

It is evident that a significantly higher proportion of parents who are willing to vaccinate their children against TBE tend to rely on sources deemed reliable. Specifically, 90.14% of the parents who chose to vaccinate their children consulted reliable sources, compared to 85.45% of those who opted not to vaccinate.

Conversely, the reliance on unreliable sources of information is less among parents who decide to vaccinate (72.59%) than those who do not (78.41%), with the difference again proving statistically significant (*p* = 0.001).

A particularly notable finding is the distinct preference for reliable sources among those who vaccinated their children, where 25.92% relied exclusively on reliable sources, in contrast to just 19.09% among those who did not vaccinate, with the difference being highly significant (*p* < 0.001).

### 3.6. Parental Perceptions and Attitudes toward Infectious Disease Danger

This section focuses on how different infectious diseases are perceived in terms of their danger to public health, and how these perceptions correlate with the willingness to vaccinate children against TBE. The analysis provides insights into the general awareness and seriousness attributed to various infectious diseases within the overall sample and stratified these perceptions by those who are willing versus unwilling to vaccinate their children against TBE.

The following barplot, Figure 3, presents the percentage of participants who perceive each listed infectious disease as dangerous, further analyzed by their willingness to vaccinate against TBE.

Starting with TBE, it is perceived as highly dangerous by an overwhelming 95.97% of the sample which was significantly higher than other diseases (*p* < 0.001). This high perception likely reflects either a well-established awareness of the disease’s severity or effective public health communications regarding its risks.

Similarly high levels of concern are observed for other serious conditions. Invasive meningococcal disease and hepatitis A virus are perceived as dangerous by 93.67% and 92.26% of respondents, respectively, indicating strong awareness of the risks associated with these diseases. Human Papillomavirus (HPV) and SARS-CoV-2, the virus responsible for COVID-19, are also perceived as highly dangerous by 88.10% and 78.15% of the sample. The slightly lower perception of danger regarding SARS-CoV-2 might reflect varying public messages and the evolving nature of the information surrounding the COVID-19 pandemic.

In contrast, diseases such as chickenpox and influenza are perceived as dangerous by 55.85% and 70.59% of the participants, respectively. The relatively lower percentage for chickenpox can be attributed to its common occurrence and typically mild symptoms in children, which may lead to its underestimation in terms of danger (see Appendix A).

The analysis of data stratified by willingness to vaccinate children against TBE, reveals significant differences between those who are willing to vaccinate and those who are not. This distinction is crucial as it highlights how risk perception directly influences health behavior, particularly vaccination decisions.

For diseases such as chickenpox, influenza, and SARS-CoV-2, there are notable disparities in perceived danger between the two groups. Those who are willing to vaccinate their children against TBE perceive these diseases as significantly more dangerous compared to those unwilling to vaccinate. Specifically, 74.91% of pro-vaccination respondents regard chickenpox as dangerous, compared to only 48.17% of those against vaccination. Similarly, for influenza, the perception of danger is 84% among those willing to vaccinate, versus 65.18% among those unwilling. The perception of the risk associated with SARS-CoV-2 also follows this trend, with 90.61% of the pro-vaccination group considering it dangerous, significantly higher than the 72.84% in the non-vaccinating group.

For more severe diseases like TBE, invasive meningococcal disease, hepatitis A, and HPV, although the overall perception of danger remains high across both groups, those willing to vaccinate consistently perceive these diseases as more dangerous. For instance, 98.62% of those willing to vaccinate perceive TBE as dangerous compared to 94.90% of those unwilling. This pattern persists across other serious diseases, indicating a strong correlation between a heightened perception of disease severity and the willingness to engage in preventative measures such as vaccination.

### 3.7. Parental Perceptions and Attitudes toward Vaccine Safety

The data contained in Figure 4 provide a detailed look into the perceptions of vaccine safety for various infectious diseases, segmented by the overall sample size that rates those vaccines as safe (ratings 4 and 5: safe and very safe) and further differentiated by the willingness to vaccinate children against TBE.

The TBE vaccine, hepatitis A vaccine, and the chickenpox vaccine exhibit the highest levels of perceived safety, with 60.96%, 62.31%, and 65.28% of respondents respectively considering them rather or very safe. These high percentages indicate a strong trust in these vaccines, which might be attributed to their long-standing availability and well-documented efficacy and safety profiles.

Closely following are the meningococcal, HPV, and influenza vaccines, which are perceived as safe by 58.82%, 58.06%, and 55.09% of respondents, respectively. The slightly lower perception of safety for the influenza vaccine could be influenced by the annual changes to the vaccine composition, which might affect public confidence as compared to vaccines that do not vary year to year. Lastly, the COVID-19 mRNA vaccine shows the lowest perceived safety, with 39.30% (see Appendix A).

For each vaccine, a higher percentage of respondents who are willing to vaccinate their children against TBE perceive the vaccines as safe compared to those who are not willing. Specifically, the perception of safety is consistently about 30–40 percentage points higher among those willing to vaccinate. For instance, 86.93% of those willing to vaccinate view the chickenpox vaccine as safe, compared to 56.02% of those who are not. This trend is seen across all vaccines, with the influenza, TBE, meningococcal, hepatitis A, HPV, and COVID-19 mRNA vaccines showing similar patterns.

#### Parental Perceptions and Attitudes toward Vaccine Safety over Time

To evaluate whether perceptions of vaccine safety changed over the duration of our study, we conducted an analysis focused only on the online questionnaires since they provided precise timestamps. The data were segmented by month, and we performed a Kruskal–Wallis test followed by Dunn’s test for pairwise comparisons.

Our analysis revealed that there were statistically significant differences in vaccine safety ratings between December 2020 and January 2021. Specifically, the median vaccine safety ratings were consistently higher in January 2021, with a median score of 4 for both the overall vaccine safety and all individual vaccines. In contrast, December 2020 had a median safety rating of 2 for both overall and all individual vaccines. This indicates that participants rated vaccines as significantly safer in January 2021 compared to December 2020.

## 4. Discussion

In Poland, the National Immunization Program (NIP) categorizes vaccines into two groups: mandatory and recommended. The mandatory vaccines include BCG (bacillus Calmette–Guérin), hepatitis B, DTP (diphtheria, tetanus, pertussis), IPV (inactivated poliovirus vaccine), Hib (Haemophilus influenzae type b), MMR (measles, mumps, rubella), PCV (pneumococcal conjugate vaccine), and the rotavirus vaccine [15].

In contrast, the recommended vaccines consist of the HPV (human papillomavirus) vaccine, the influenza vaccine, meningococcal vaccines, the varicella (chickenpox) vaccine, hepatitis A, and the TBE vaccine, which is particularly advised for individuals living in or traveling to areas with high tick activity [15].

Our study included the TBE vaccine in particular because our hospital is located in an area with abundant forests and a high number of trees. This environment attracts many people for outdoor activities, such as mushroom-picking or jogging, increasing their risk of exposure to ticks and the potential for contracting TBE.

Overall, we can conclude that the general willingness in Poland to vaccinate one’s child against TBE is relatively low at 28.87%. Despite the high incidence of severe infections with TBEV, Poland demonstrates low vaccination rates, possibly influenced by or in conjunction with low willingness [12]. According to data from the Polish National Institute of Public Health, the TBE vaccine was given to approximately 1.1% of the entire Polish population between 2011 and 2020 [12].

With regards to sociodemographic factors, our study found that neither gender nor educational background are statistically significant predictors of TBE vaccination willingness. When it comes to age, we did see a higher vaccinate among very young parents (up to 19 years old) and those between 50–59 years. This could reflect a heightened awareness or concern about TBE in these groups, possibly due to different life stages and associated health priorities. In contrary to other age groups which do not show significant differences, implying that age is not a universal influencer in TBE vaccination decisions for the majority of parents. However, this variable in prior studies revealed varying outcomes. Research in Switzerland and Austria showed higher TBE vaccination rates in individuals under 60 years old [16], while in Sweden and northeastern Poland, those over 60 had higher uptake compared to younger adults and children [12,17]. These opposing findings combined with the fact that there was no significant difference in other age groups in our study imply that age may not always be a reliable determinant of TBE vaccination behavior. Discrepancies are likely tied to occupation or leisure preferences, differing across countries. Additionally, the observation that larger cities with a population exceeding 300,000 inhabitants demonstrate a higher willingness to vaccinate against TBE might hint at an urban–rural divide in vaccination attitudes. This disparity in willingness to vaccinate based on rural settings aligns with findings from the study conducted by Riccò et al. (2019), which focused on farmers in northeastern Italy [18]. The data suggest that the context of urban living, with its unique infrastructure, healthcare access, and possibly higher awareness of communicable diseases, may contribute to a more favorable attitude towards TBE vaccination in urban populations. Understanding these nuances in vaccination behavior in different settings is crucial for targeted public health interventions to ensure widespread protection against tick-borne diseases like TBE. Furthermore, our data indicate that parents without children are significantly more inclined to vaccinate their future potential child, whereas those with three or more children exhibit a diminished willingness to vaccinate their children. While the literature on this topic is quite limited, a study conducted in India on predictors of parents’ willingness to vaccinate their children against COVID-19 indicated that parents with only one or two children showed a higher likelihood of vaccination compared to others [19]. Lastly, both our study as well as previous studies on the same topic see a positive relationship between willingness to vaccinate against TBE whether or not the participant lives in a TBE-endemic area. For example, there appears to be a north–south gradient in Germany regarding TBE vaccination rates, with the highest rates observed in the two major endemic areas, which are the southern federal states of Baden-Württemberg and Bavaria [20,21]. In another study, one of the factors positively associated with TBE vaccination included residency in high-risk areas [22]. These findings align with the health belief model, which—among other findings—suggests that individuals are more likely to engage in preventive behaviors, such as vaccination, when they perceive a higher risk of contracting the disease [23]. In conclusion, understanding the geographical variations in TBE occurrence can inform targeted public health strategies to address regional disparities and promote broader vaccine coverage in high-risk and/or low vaccination rate areas.

A large portion of parents (72.57%) support child vaccination, correlating with 95.98% of supporters of child vaccinations also being willing to vaccinate against TBE, contrasting with only 63.11% of non-supporters of TBE vaccination. When it comes to parental knowledge about vaccines a few things crystallized. The majority of respondents (68.53%) believe that not vaccinating children poses a greater risk than vaccinating them. Fear of disease consequences proves to be a more significant driver of TBE vaccination acceptance than fear of vaccine-induced harm. Additionally, only a small proportion (27.62%) of participants prefer natural immunity methods over vaccinations. These observations suggest a high level of trust in vaccination efficacy and demonstrate a solid grasp of the benefits of vaccinations. Furthermore, most parents (82.39%) who favor vaccines over natural immunity are willing to vaccinate their kid against TBE compared to minority (4.60%) of respondents who prioritize natural methods. This indicates a strong belief in the efficacy of vaccinations over natural methods among those choosing to vaccinate.

However, concerns about autism risks associated with vaccines persist (27.10%) despite scientific evidence refuting this connection. Moreover, misinformation linking vaccines to autism poses a hurdle in vaccination efforts, decreasing the willingness to vaccinate against TBE. In general, we can say that a lack of knowledge on and misinformation about vaccines have always been a clear barrier to willingness to vaccinate. Multiple studies done found that on one hand, inadequate knowledge of tick-borne diseases contributed to poor adoption of preventive measures [18,22], and on the other hand, people who had better knowledge of tick-borne diseases were more likely to be vaccinated against TBE [18].

Besides that, the underestimated impact of decreasing or covering vaccine costs is worth noting. Our data show that 57.64% would opt for combined vaccines if they were freely available. In one study, while cost was cited as a barrier by some travelers to TBE-endemic areas, healthcare providers perceived cost as a more significant deterrent to vaccination [24]. Generally speaking, low-income households face a higher barrier to TBE vaccination due to the cost of the vaccine, leading to lower vaccination rates among this group [21,25,26]. In conclusion, the data underscore the substantial influence of reducing or subsidizing vaccine costs on vaccination decision-making, highlighting the critical role of addressing financial barriers in promoting immunization rates.

Compared to other non-mandatory childhood vaccines, the willingness to vaccinate against TBE falls between the higher acceptance rates for vaccines like chickenpox and meningococcal disease, and the lower acceptance rates for vaccines such as influenza, HAV, and HPV. An important discovery is that a significant correlation exists between the willingness to vaccinate against TBE and other non-mandatory vaccines. This finding sheds light on the interconnected nature of vaccine decision-making processes, suggesting that individuals who demonstrate openness to vaccination against TBE might also be more inclined to consider and accept other non-mandatory vaccines.

More reliable sources like healthcare professionals and authoritative health organizations (WHO, CDC) play a significant role in influencing parents’ decisions to vaccinate against TBE, with willing parents more likely to consult them. Moreover, exclusive reliance on reliable sources correlates strongly with the decision to vaccinate. Conversely, parents unwilling to vaccinate rely more on the Internet. This may indicate that the Internet, while a rich source of information, might also be a platform for spreading misinformation or conflicting information that could influence decisions against vaccination. However, it is crucial to understand that correlation does not imply causation. While our study focused on identifying the sources of vaccine knowledge used by participants, we did not directly test the impact of the internet on attitudes and perceptions towards vaccines. Nevertheless, numerous past studies have demonstrated that internet usage can have a detrimental effect on individuals’ knowledge, attitudes, and perception regarding vaccines. The internet’s potential to propagate misinformation and present conflicting information poses a risk of influencing decisions against vaccination [27,28]. Interestingly and unexpectedly, individuals who turn to popular science books and guides for information on vaccines exhibit higher hesitancy towards the TBE vaccine, as a larger percentage of them refuse rather than accept the vaccine. However, the causal relationship here remains unknown as well, leading to a “chicken-and-egg” dilemma. It is unclear if popular science books contribute to anti-vaccine attitudes or if those with anti-vaccine attitudes are inclined to read more popular science books. Further investigation is necessary to understand the association between reliance on popular science books and vaccine hesitancy. In conclusion, the influence of healthcare professionals and credible health organizations remains vital in promoting vaccination and combating misinformation in parental decision-making.

As far as TBE disease ratings are concerned, it is perceived as highly dangerous by a large majority of participants, with a significantly higher perception of danger compared to other diseases. Diseases like invasive meningococcal disease, hepatitis A, HPV, and COVID-19 are also perceived as dangerous by a significant portion of respondents. Diseases, such as chickenpox and influenza, are perceived as dangerous by a considerable percentage of participants, albeit to a lesser extent. This is important because infectious disease danger perception affects vaccine willingness. Those willing to vaccinate against TBE generally perceive all infectious diseases as more dangerous than those not willing to vaccinate. Thus, we can state that higher perceptions of disease danger are associated with a greater willingness to engage in preventative measures, such as vaccination, highlighting the important role of risk perception in shaping health behavior. Past studies come to the same conclusion. The level of risk perception regarding an infectious disease is a crucial factor influencing people’s willingness to get vaccinated. Individuals who perceive a high risk of contracting the disease and view its consequences as severe are more inclined to receive vaccinations [17,29,30,31,32,33,34,35,36]. In this context, it is also worth mentioning that people who have previously experienced a tick bite are statistically more willing to be vaccinated against TBE, probably for the same reason. Their prior negative encounter with the infectious disease may have predisposed them to perceive it as significantly more dangerous in accordance with the health belief model [23], which in fact increased the likelihood of them being vaccinated in the future [12,18,22].

A significant percentage of respondents perceive vaccines for TBE, hepatitis A, and chickenpox to be highly safe, indicating strong trust in these vaccines due to their proven efficacy and safety records. Similarly, meningococcal, HPV, and influenza vaccines are generally considered safe, with the influenza vaccine possibly having slightly lower perceived safety due to annual composition changes in its composition. Notably, there is a consistent trend where more respondents willing to vaccinate against TBE view the vaccines as safe compared to the unwilling group, reflecting a broader confidence in vaccines and trust in medical institutions. This pattern is observed across all vaccines, with a significant difference in the perception of safety between the two groups. Thus, inadequate risk perception and irrational fears regarding TBE vaccine are identified as negative predictive variables for vaccination [22,32]. The data emphasizes the crucial role of perceived safety in parental decisions regarding vaccinating their children and highlights the necessity for continuous communication to address concerns and foster trust in vaccination practices. Establishing trust, promoting transparency, and providing education are inevitable aspects for doctors in enhancing vaccine confidence.

The COVID-19 mRNA vaccine, however, is perceived to have the lowest safety among all mentioned vaccines. This is likely due to its novelty, rapid development during the pandemic, and associated hesitancy [19,29]. Another factor might be the excessive media coverage of COVID-19 and its vaccine during the pandemic. Vaccines that receive extensive media attention and are heavily debated in public are paradoxically more susceptible to populist attitudes. Then, individuals with higher levels of science-related populism in turn have lower vaccination confidence and feel less obligated to contribute to collective health benefits through vaccination [37].

In summary, we can say that educating the public and implementing robust vaccination campaigns are pivotal in increasing vaccine confidence by shedding light on the danger of infectious diseases and the safety of vaccines and combating misinformation. The Austrian vaccination paradox from the past few decades showed us how important proper vaccination campaigns are at increasing vaccine uptakes. Today, Austria has achieved the highest TBE vaccination rate globally (82%) through consistent social marketing, effective vaccination campaigns, effective vaccines, and a decrease in TBE cases. On the contrary, while the influenza vaccine received a stronger vaccination recommendation compared to the TBE vaccine, its coverage and uptake were lower (8%, one of the lowest rates worldwide), possibly due to ineffective execution or other factors [38].

This study has several limitations that must be acknowledged, which could affect the generalizability and interpretation of the results.

One primary limitation is the reliance on self-reported data, which can introduce response bias. Participants may provide socially desirable answers or may not accurately recall their vaccination decisions and attitudes. This limitation is commonly noted in survey-based research, as highlighted in a previous study in Italy from 2020, which discussed similar concerns in their study on occupational physicians’ attitudes towards TBE vaccination [39].

A significant selection bias cannot be ruled out here either. Participating proactively could be due to a certain extreme or passionate attitude or opinion about vaccinations, which might be enhanced in online surveys due to the perk of anonymity.

The sample also predominantly consists of mothers, accounting for over 90% of the respondents. This gender imbalance might limit the generalizability of the findings to all guardians, as fathers’ attitudes towards vaccination might differ.

The paper surveys were conducted in a specific region of Poland, particularly focusing on an endemic area in northeastern Poland. This regional focus means the findings may not reflect the attitudes of guardians in other parts of the country or in regions with different epidemiological profiles for TBE. Comparatively, Kunze and Kunze (2015) discussed the regional variability in vaccination uptake and attitudes within Austria, highlighting how local epidemiology can influence behavior [38].

Lastly, while observed correlations suggest a link between vaccine safety rating and knowledge gaps and general attitude towards vaccines, causation cannot be conclusively established due to potential undiscovered factors. A previous study analyzing “Knowledge, Attitudes, and Behaviors Regarding Lyme Borreliosis Prevention in the Endemic Area of Northeastern Poland” came to a similar conclusion [11].

Lastly, this study did not use a multivariable model to adjust for the confounding variables that could have given a more robust analysis. As such, an analysis that might account for potential confounders was not conducted, and this might affect the validity of the observed associations. Future studies need to apply multivariable models for the adjustment of these confounding factors to better understand the relationships among variables.

The robustness of this study is underscored by its extensive participant base. With a comprehensive dataset derived from 3030 surveys, we can assert with confidence that the findings carry significant weight and reflect a reliable cross-section of attitudes towards the TBE vaccine. The data elucidate key knowledge gaps and sources of vaccine knowledge that, if addressed through targeted educational campaigns, could significantly enhance public health strategies.

Future investigations should consider a comparative analysis of paper versus online questionnaires to discern any differences in attitudes towards TBE vaccines. This comparison may reveal whether certain biases, perspectives and attitudes are overrepresented in one format over the other, which could provide invaluable insights for the design of future epidemiological studies.

Furthermore, it would be prudent to examine the relationship between vaccine safety and infectious disease danger ratings by performing correlation testing. Employing certain statistical learning and machine learning tools could offer valuable insights into which model best predicts vaccine safety ratings using infectious disease danger ratings as a predictor. This could be used to reveal a possible relationship between the two, in which case we could target two points of attack in increasing vaccination rates.

## 5. Conclusions

This cross-sectional survey study represents the first of its kind to extensively examine attitudes towards the TBE vaccine and compare them to TBE and other infectious diseases and their vaccines in Poland. The fact that it is conducted with such a large sample size consolidates its impact.

Myriad factors influence the willingness of parents to vaccinate their children against TBE in Poland. It is important to address knowledge gaps and dispel misinformation to enhance vaccine coverage. Furthermore, the influence of healthcare professionals and reliable sources in shaping vaccination decisions is pivotal.

Risk perception of infectious diseases and vaccines are key determinants in vaccination behavior. There is a critical need for targeted education initiatives to increase vaccine confidence, ultimately promoting better vaccine uptake and decreasing TBE cases.

## Figures and Tables

**Figure 1 vaccines-12-00918-f001:**
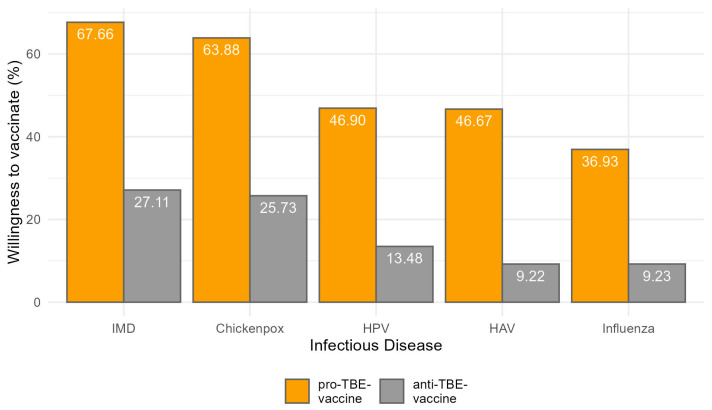
Percentage of willingness to receive specific non-mandatory childhood vaccinations among pro- and anti-TBE-vaxxers.

**Figure 2 vaccines-12-00918-f002:**
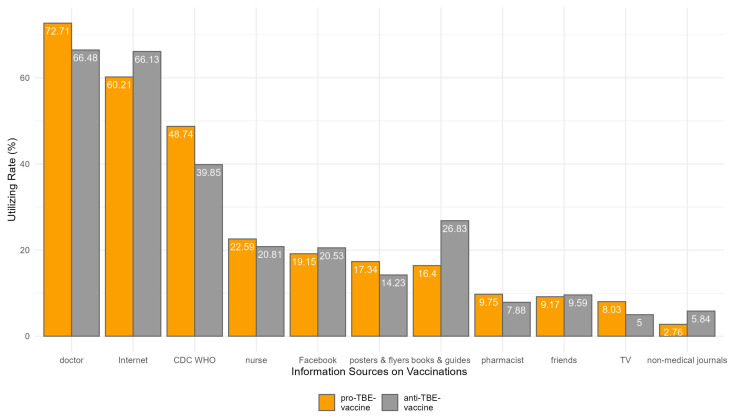
Percentages of utilized information source on vaccines among pro- and anti-TBE-vaccine participants.

**Figure 3 vaccines-12-00918-f003:**
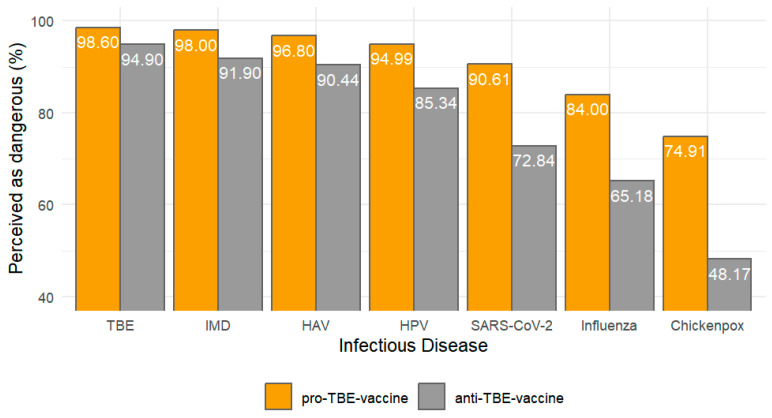
Percentage of infectious diseases perceived as dangerous among pro- and anti-TBE-vaccine participants.

**Figure 4 vaccines-12-00918-f004:**
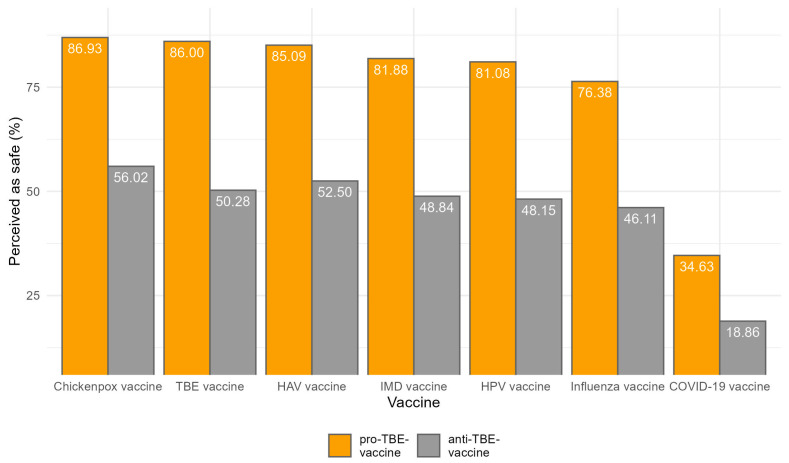
Percentage of vaccines perceived as safe among pro- and anti-TBE-vaccine participants.

**Table 1 vaccines-12-00918-t001:** Demographic profile of parents and their intent to vaccinate their children against TBE.

Characteristic	N	Overall Sample ^a^	Willingness to Vaccinate Children against TBE		*p*
			Yes, n = 872 ^a^	No, n = 2158 ^a^	
Gender:	3024				0.087 ^b^
female		2759 (91.24%)	779 (89.85%)	1980 (91.79%)	
male		265 (8.76%)	88 (10.15%)	177 (8.21%)	
Age:	3021				
up to 19 years		5 (0.17%)	4 (0.46%)	1 (0.05%)	**0.027** ^c^
20–29 years		676 (22.38%)	193 (22.29%)	483 (22.41%)	0.925 ^d^
30–39 years		1814 (60.05%)	507 (58.55%)	1307 (60.65%)	0.218 ^d^
40–49 years		470 (15.56%)	141 (16.28%)	329 (15.27%)	0.525 ^d^
50–59 years		45 (1.49%)	19 (2.19%)	26 (1.21%)	**0.045** ^d^
60–69 years		9 (0.30%)	2 (0.23%)	7 (0.32%)	1.000 ^c^
70 years or above		2 (0.07%)	0 (0%)	2 (0.09%)	1.000 ^c^
Education:	3023				0.355 ^b^
high		2360 (78.07%)	687 (79.33%)	1673 (77.56%)	
vocational		112 (3.70%)	35 (4.04%)	77 (3.57%)	
secondary		534 (17.66%)	138 (15.94%)	396 (18.36%)	
primary		17 (0.56%)	6 (0.69%)	11 (0.51%)	
Locality:	3017				**0.004** ^b^
big city (over 300,000 inhabitants)		1072 (35.53%)	350 (40.51%)	722 (33.53%)	**<0.001** ^d^
city (up to 300,000 inhabitants)		805 (26.68%)	214 (24.77%)	591 (27.45%)	0.108 ^d^
town (up to 30,000 inhabitants)		531 (17.60%)	140 (16.20%)	391 (18.16%)	0.176 ^d^
village		609 (20.19%)	160 (18.52%)	449 (20.85%)	0.126 ^d^
Number of children:	3012				**<0.001** ^b^
No children		69 (2.29%)	46 (5.33%)	23 (1.07%)	**<0.001** ^d^
1–2 children		2354 (78.15%)	691 (80.07%)	1663 (77.38%)	0.192 ^d^
3 or above children		589 (19.56%)	126 (14.60%)	463 (21.54%)	**<0.001** ^d^
Voivodeship endemic for TBE	3030	938 (30.96%)	308 (35.32%)	630 (29.19%)	**0.001** ^b^

^a^ n (%); ^b^ Pearson’s chi-squared test; ^c^ Fisher’s exact test; ^d^ proportion test; bold *p*-values denote statistical significance.

**Table 2 vaccines-12-00918-t002:** Comparative analysis of general attitudes towards vaccination and willingness to vaccinate children against TBE.

Characteristic (Question)	N	Overall Sample ^a^	Willingness to Vaccinate Children against TBE		*p*
			Yes, n = 872 ^a^	No, n = 2158 ^a^	
Q1. Do you think that children should be vaccinated?	3029				**<0.001** ^b^
yes		2198 (72.57%)	836 (95.98%)	1362 (63.11%)	**<0.001** ^c^
no		663 (21.89%)	15 (1.72%)	648 (30.03%)	**<0.001** ^c^
don’t know		168 (5.55%)	20 (2.30%)	148 (6.86%)	**<0.001** ^c^
Q2. Which option regarding vaccination of children is more dangerous?	3028				**<0.001** ^b^
vaccination of children		829 (27.38%)	48 (5.52%)	781 (36.19%)	**<0.001** ^c^
lack of vaccination of children		2075 (68.53%)	798 (91.72%)	1277 (59.18%)	**<0.001** ^c^
don’t know		124 (4.10%)	24 (2.76%)	100 (4.63%)	**0.018** ^c^
Q4. Do you believe that natural methods of boosting immunity are more effective than vaccinations in preventing infections?	3023				**<0.001** ^b^
yes		835 (27.62%)	40 (4.60%)	795 (36.91%)	**<0.001** ^c^
no		1718 (56.83%)	716 (82.39%)	1002 (46.52%)	**<0.001** ^c^
don’t know		470 (15.55%)	113 (13%)	357 (16.57%)	**0.014 ^c^**
Q6. If you had the opportunity to use high combination vaccines for free, would you choose to give them to your child?	3029				**<0.001** ^b^
yes		1746 (57.64%)	711 (81.63%)	1035 (47.96%)	**<0.001** ^c^
no		1062 (35.06%)	93 (10.68%)	969 (44.90%)	**<0.001** ^c^
don’t know		221 (7.30%)	67 (7.69%)	154 (7.14%)	0.600 ^c^
Q9. In your opinion, can vaccinations cause autism?	3026				**<0.001** ^b^
yes		820 (27.10%)	66 (7.58%)	754 (34.99%)	**<0.001** ^c^
no		1429 (47.22%)	622 (71.41%)	807 (37.45%)	**<0.001** ^c^
don’t know		777 (25.68%)	183 (21.01%)	594 (27.56%)	**<0.001** ^c^

^a^ n (%); ^b^ Pearson’s chi-squared test; ^c^ proportion test; bold *p*-values denote statistical significance.

**Table 3 vaccines-12-00918-t003:** Comparative analysis of the sources in terms of reliability with stratifying by the willingness to vaccinate children against TBE.

Characteristic	N	Overall Sample	Willingness to Vaccinate Children against TBE		*p* ^b^
			Yes, n = 872 ^a^	No, n = 2158 ^a^	
Reliable source of information	3030	2630 (86.80%)	786 (90.14%)	1844 (85.45%)	**0.001**
Unreliable sources of information	3030	2325 (76.73%)	633 (72.59%)	1692 (78.41%)	**0.001**
Reliable sources only	3030	638 (21.06%)	226 (25.92%)	412 (19.09%)	**<0.001**

^a^ n (%); ^b^ Pearson’s chi-squared test; bold *p*-values denote statistical significance.

## Data Availability

The raw data supporting the conclusions of this article will be made available by the corresponding authors on reasonable request.

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
