# Peer review of "Attitudes towards the Tick-Borne Encephalitis Vaccine among Children’s Guardians: A Cross-Sectional Survey Study in Poland"

_vaccines, 2024, doi:10.3390/vaccines12080918_

Round 1
Reviewer 1 Report
Comments and Suggestions for Authors
Dear authors,
I find this research highly interesting and well written.
I have only one concern and it is related to presentation of results as is it really easy to get lost in this section.
Please consider to switch tables with charts where you will highlight p values. Other information that is now in the tables can be transferred to supplementary files. In that way you will increase readability without loss of data that you want to present.
Author Response
Comments: “Please consider to switch tables with charts where you will highlight p values. Other information that is now in the tables can be transferred to supplementary files. In that way you will increase readability without loss of data that you want to present.”
Response: Thank you very much for taking the time to review our manuscript and also pointing out this issue. We agree that too many tables make it hard to grasp the results chapter. This is why we deleted tables 3 and 6 and included figures 1 and 2, respectively, which communicate the same idea. Furthermore, we also swapped tables 4 and 7 for grouped bar plots to increase readability without loss of the data we want to convey.
Reviewer 2 Report
Comments and Suggestions for Authors
This is a well made cross-sectional survey about the attitude of polish people towards vaccination. While its results may be predictable, they offer insight about vaccine hesitancy and possible ways to overcome it.
My main complaint about the article is that the results are presented and then discussed in the results section, only to be discussed again in the discussion section: this may be my personal preference, but I think that data should be presented in the briefest way possible in the result section and then discussed after. I find the article as it is, and, in particular, the result section, very dispersive and difficult to read, and there is a marked redundancy between results and discussion. E.g: lines 132-136 should be in discussion; same for lines 140-141, 150-152 etc.
I have some other minor considerations:
- A copy of the questionnaire should be added to supplementary matherials.
- I would confront the characteristics of the study population with the general characteristics of the polish population to investigate the reproducibility of results.
- Why the authors did not analyze data using a multivariable model to adjust for confounding factors? Please explain in the limitations section.
- Fig 1: I would not use the term "pro-vaxxers" or "anti-vaxxers" due to the "political" significance they acquired during COVID19: I wouldn't define a vaccine hesitant person as an anti-vaxxer (all anti-vaxxers are vaccine-hesitant, not every vaccine hesitant is an anti-vaxxer).
- Please move to the material and methods section the criteria you used to divide between reliable and non-reliable sources (lines 292-298).
- Lines 338-339: i wouldn't say that the medical community consider as "less severe diseases" influenza and chickenpox. Please provide an appropriate reference or rephrase.
- The "strenght of study" section is maybe a little bit "strong": I do agree that the number of participants is more than satisfactory, but, still, some of the demographics are skewed: 90% of respondents were women (are mothers the only ones who take decisions regarding vaccinations?), a really high proportion of respondents have a high level of education (what is the average level of education in Poland?): see also the first point of this bullet-list.
- This may be also personal preference, but I prefer the discussion divided in discussion, limitations and conclusions without other sub-paragraphs.
Comments on the Quality of English Language
An English revision may be helpful for fluency, but the article is comprensible and grammatically correct (for a non-mother-tongue speaker at least).
Author Response
Comment 1: My main complaint about the article is that the results are presented and then discussed in the results section, only to be discussed again in the discussion section: this may be my personal preference, but I think that data should be presented in the briefest way possible in the result section and then discussed after. I find the article as it is, and, in particular, the result section, very dispersive and difficult to read, and there is a marked redundancy between results and discussion. E.g: lines 132-136 should be in discussion; same for lines 140-141, 150-152 etc.
Response 1: Thank you for pointing this out, we agree with this comment therefore we moved and modified lines 133-137 which are now lines 401-405, as well as lines 189-191 which are now 449-452, 272-275 which are now 483-486, and lines 313,314 which are now 559-560.
Lines 139-141, 150-152, 183,184, 227-230, 241-243, 262-264, 267-269, 279-280, 285-287, 307,308, 375-377, 379-382, 384-390, 398-400 got deleted as this subject has been already discussed in discussion section.
Comment 2 : A copy of the questionnaire should be added to supplementary materials.
Response 2: Thank you for pointing this out, we agree with this comment therefore we added a copy of the questionnaire to supplementary materials.
Comment 3: I would confront the characteristics of the study population with the general characteristics of the Polish population to investigate the reproducibility of results.
Response 3: Thank you for pointing this out, we agree with this comment therefore we Added a paragraph (lines 147-150) about general characteristics of the polish population.
Comment 4: Why the authors did not analyze data using a multivariable model to adjust for confounding factors? Please explain in the limitations section.
Response 4: We agree with this comment, therefore we addressed this issue in the limitations section of the discussion chapter, noting that the use of a multivariable model could be a valuable approach for future research.
Comment 5: Fig 1: I would not use the term "pro-vaxxers" or "anti-vaxxers" due to the "political" significance they acquired during COVID19: I wouldn't define a vaccine hesitant person as an anti-vaxxer (all anti-vaxxers are vaccine-hesitant, not every vaccine hesitant is an anti-vaxxer).
Response 5: Thank you for pointing this out, we agree with this comment therefore in line 207 term pro-TBE-vaccers got changed into parents willing to vaccinate against TBE as well as term anti-TBE-vaccer in line 208 into parents unwilling to vaccinate against TBE.
Comment 6: Please move to the material and methods section the criteria you used to divide between reliable and non-reliable sources (lines 292-298).
Response 6: Thank you for pointing this out, we agree with this comment therefore lines 291-298 were modified and moved to Materials and Methods section (lines 95-102).
Comment 7: Lines 338-339: i wouldn't say that the medical community consider as "less severe diseases" influenza and chickenpox. Please provide an appropriate reference or rephrase.
Response 7: Thank you for pointing this out, we agree with this comment therefore lines 334 and 507 got rephrased from “In contrast, diseases typically considered less severe by the medical community” to “In contrast, diseases such as chickenpox and influenza”.
Comment 8: The "strenght of study" section is maybe a little bit "strong": I do agree that the number of participants is more than satisfactory, but, still, some of the demographics are skewed: 90% of respondents were women (are mothers the only ones who take decisions regarding vaccinations?), a really high proportion of respondents have a high level of education (what is the average level of education in Poland?): see also the first point of this bullet-list.
Response 8: Thank you for pointing this out, we agree with this comment therefore we added a paragraph (lines lines 147-150) about general characteristics of the polish population which our respondents represent closely but not directly.
Comment 9: This may be also personal preference, but I prefer the discussion divided in discussion, limitations and conclusions without other sub-paragraphs.
Response 9: Thank you for pointing this out, we agree with this comment therefore we divided discussion in discussion, limitations and conclusions.
Reviewer 3 Report
Comments and Suggestions for Authors
It would be of great importance also to comment of the existing vaccines on the NIP in Poland.
Could the authors add some comments regarding the coverage rate of the vaccines that are on the NIP along with some comments if vaccination mandatory or not in the country
Could the authors why did they choose this particular vaccine and no other
Could the authors comment if the attitude towards vaccine changed in time
Author Response
Comment 1: “It would be of great importance also to comment of the existing vaccines on the NIP in Poland.”
Response 1: Thank you for pointing this out, we agree with this comment therefore we included a quick overview on this topic in the discussion section.
Comment 2: “Could the authors add some comments regarding the coverage rate of the vaccines that are on the NIP along with some comments if vaccination mandatory or not in the country”
Response 2: Thank you for pointing this out, we agree with this comment therefore we added a paragraph in the introduction section about this topic. We were unable to allocate official data from the NIPH–NIH regarding the coverage rates of the non-mandatory vaccines included in our study. Therefore, we have provided absolute numbers and the quantity of doses administered for each vaccine in 2022.
Comment 3: “Could the authors why did they choose this particular vaccine and no other”
Response 3: Thank you for pointing this out, in the discussion section we clarified again in detail, why we chose TBE vaccine as our main target research gap.
Comment 4: “Could the authors comment if the attitude towards vaccine changed in time”
Response 4: This is a very interesting question and something we wanted to find out as well. Thus, we took all online questionnaires since they had precise timestamps and grouped them by month. We found out that participants rated vaccines as safer in January 2021 compared to December 2020. All that we wrote down in our revised manuscript in lines 393-404.
Round 2
Reviewer 1 Report
Comments and Suggestions for Authors
Dear Authors,
Thank you for responding to my comment.